# LPRules: Rule Induction in Knowledge Graphs Using Linear Programming

## Abstract

Knowledge graph (KG) completion is a well-studied problem in AI. Rule-based methods and embedding-based methods form two of the solution techniques. Rule-based methods learn first-order logic rules that capture existing facts in an input graph and then use these rules to reason about missing facts. A drawback of such methods is the lack of scalability to large datasets. We present a simple linear programming (LP) model to choose rules from a list of candidate rules and assign weights to them. For smaller KGs, we use simple heuristics to create the candidate list. For larger KGs, we start with a small initial candidate list, and then use column generation to add more rules in order to improve the LP model objective value. To foster interpretability and generalizability, we limit the complexity of the set of chosen rules via explicit constraints, and tune the complexity hyperparameter for individual datasets. We show that our method can obtain state-of-the-art results for three out of four widely used KG datasets, while generating compact rule sets and taking significantly less computing time than other popular rule learners, including some based on neuro-symbolic methods. The improved scalability of our method allows us to tackle large datasets such as YAGO3-10.

## 1 Introduction

Knowledge graphs (KG) are used to represent a collection of known facts via labeled directed edges. Each node of the graph represents an *entity*, and a labeled directed edge from one node to another indicates that the pair of nodes satisfies a binary relation given by the edge label. A *fact* in the knowledge graph is a triplet of the form $(a, r, b)$ where $a$ and $b$ are nodes, and $r$ is a binary relation labeling a directed edge from $a$ to $b$ indicating that $r(a, b)$ is true. Consider a KG where the nodes correspond to distinct cities, states, and countries and the relations are one of *capital_of*, *shares_border_with*, or *part_of*. A fact *(a, part_of, b)* in such a graph corresponds to a directed edge from *a* to *b* labeled by *part_of* implying that *a* is a part of *b*. Practical knowledge graphs are often incomplete (they do not contain all true representable facts) and noisy (they can have inconsistencies or errors). Knowledge graph completion (KGC), triple classification, entity recognition, and relation extraction are common tasks for extracting implied information from KGs. See Ji et al. (2021) for a recent survey on KGs. KGC involves using known facts in a KG to infer additional (missing) facts. One approach for KGC is to learn first-order logic rules that encode known facts.

In the example above, we could learn a "rule" *capital_of(X,Y) → part_of(X,Y)* where *X, Y* are variables that take on entity values. Then if we find a pair of entities *P,Q* such that *(P, capital_of, Q)* is a fact in the graph, we could infer that *(P, part_of, Q)* is also true and augment the set of facts with this new fact if not originally present. A more complex rule of *length* two is *capital_of(X,Y) and part_of(Y,Z) → part_of(X,Z)*. Again, applying it to entities *P* and *Q*, if there exists a third entity *R* such that *capital_of(P,R)* is a fact in the graph, and so is *part_of(R,Q)*, then we infer that *P* is a part of *Q*. KG link prediction deals with finding answers to queries of the form *part_of(P, ?)*, and we focus on finding first-order logic rules of the type above for this task. Instead of learning one rule for a relation, it is common to learn a set of rules along with rule weights, where larger weights indicate more important or more certain rules. Dealing with uncertainty or noise is essential for KG reasoning.

Learning logic rules is a well-studied area. In a paper (Lao & Cohen, 2010) on path ranking algorithms and another (Richardson & Domingos, 2006) on Markov logic networks, candidate logic rules are obtained via relational path enumeration, and then rule weights are calculated. Yang et al. (2017)

use neural logic programming to simultaneously obtain sets of rules and the weights of individual rules. Qu et al. (2021), separately find rules and rule weights, but add a feedback loop from the latter learning problem to the former. The use of recursive neural networks (RNN) to learn rules is common nowadays, though traditional rule-mining approaches remain popular (Meilicke et al., 2019).

Embedding-based methods for KGC consist of representing nodes by vectors, and relations by vector transformations that are consistent with the facts in the knowledge graph. They exhibit better scaling with KG size, and yield more accurate predictions. See the surveys by Ji et al. (2021) and Wang et al. (2017). On the other hand, rule-based methods can yield more interpretable explanations of associated reasoning steps (Qu & Tang, 2019), especially if one obtains compact rule sets (with few rules and few relations per rule). Furthermore, entity-independent rules are more generalizeable (Teru et al., 2020) and can be applied in an *inductive setting*: they can be applied to entities not considered in the learning process. Embedding-based methods work mostly in a *transductive setting*.

We propose a novel approach to learning entity-independent, weighted first-order logic rules for knowledge graph reasoning. Our approach combines rule enumeration with linear programming (LP), and completely avoids the solution of difficult nonconvex optimizaton models inherent in training RNNs. We describe a linear programming (LP) formulation with exponentially many variables corresponding to first-order rules and associated weights. Nonzero variable values correspond to chosen rules and their weights. We deal with the exponential number of variables/rules by column generation ideas from linear optimization, where we start off with some small initial set of rules and associated variables, find the best subset of these and associated weights via the partial LP defined on the initial set of rules, and then generate new rules which can best augment the existing set of rules. Our final output rule-based scoring functions resemble those in NeuralLP (Yang et al., 2017), DRUM (Sadeghian et al., 2019), and RNNLogic (Qu et al., 2021) in that we form a linear combination of rule scores (calculated differently from the above papers). As in RNNLogic, our iterative process of adding new rules is influenced by previous rules.

Our algorithm has better scaling with KG size than a many existing rule-based methods. In addition, we obtain state-of-the-art results on three out of four standard KG datasets while taking significantly less running time than some competing codes. Furthermore, we are able to obtain results of reasonable quality – when compared to embedding-based methods – on YAGO3-10, a large dataset which is difficult for existing rule-based methods. As an important goal of rule-based methods is to obtain interpretable solutions, we promote interpretability by adding a constraint in our LP formulation limiting the complexity of the chosen rule set (and this hyperparameter is tuned per dataset). In some cases, we obtain more accurate results (higher MRR) for the same level of complexity as other codes, and less complex rules for the same level of accuracy in other cases.

## 2 RELATED WORK

### 2.1 RULE-BASED METHODS

There is a rich body of literature on rule-based methods for knowledge graph reasoning. The motivation for learning rules is that they form an explicit symbolic representation of existing knowledge and are amenable to inspection and verification. Further, compact rule sets are interpretable, an efficient way to store knowledge, and useful for transfer learning. For KG applications, when rules are entity-independent, they can be used in an inductive setting (Yang et al. (2017),Teru et al. (2020)).

**Inductive Logic Programming (ILP).** In this approach, one takes as input positive and negative examples and learns logic programs that entail all positive examples and none of the negative examples. See for example Cropper & Muggleton (2016) and Cropper & Morel (2020). First-order Logic (FOL) programs in the form of a collection of chain-like Horn clauses are a popular output format. Negative examples are not available in typical knowledge graphs, and some of the positive examples can be mutually inconsistent. Evans & Grefenstette (2018) developed a differential ILP framework to generate rules for noisy data.

**Statistical Relational Learning (SRL).** SRL aims to learn FOL formulas from data and to quantify their uncertainty. Markov logic, which is a probabilistic extension of FOL, is a popular framework for SRL. In this framework, one learns a set of weighted FOL formulas; see, for example, Kok & Domingos (2005) where beam search is used to find a set of FOL rules, and rule weights are learned

via standard numerical methods. In knowledge graph reasoning, *chain-like* rules which correspond to relational paths (and to chain-like Horn clauses) are widely studied. In Lao & Cohen (2010), a weighted linear combination of rule-based functions (e.g., the function could return a probability that the rule implies a link between a pair of entities) is used as a scoring function for KG link completion. An initial set of rules is created and the weights are obtained via regression. A recent, bottom-up rule-learning algorithm with excellent predictive performance is AnyBURL (Meilicke et al., 2019). We learn FOL rules corresponding to relational paths and rule weights; our scoring function, and learning model/algorithm are different from prior work.

The column generation aspect of our work has similarities to cutting plane inference methods (Riedel, 2008; Noessner et al., 2013) for MAP inference in SRL. While we deal with an exponential number of possible rules/Horn clauses via column generation, the above papers use constraint generation/cutting plane ideas to deal with an exponential number of constraints corresponding to ground clauses.

**Neuro-symbolic methods.** In NeuralLP (Yang et al., 2017), rules and rule weights are learned simultaneously by training an appropriate RNN. Further improvements in this paradigm can be found in DRUM (Sadeghian et al. (2019)). Another neuro-symbolic approach is implemented in the Neural Theorem Prover (NTP, Rochstätel & Riedel (2017)). More general rules (than the chain-like rules found in NeuralLP) are obtained in NLIL (Yang & Song) along with better scaling behavior. Simultaneously solving for rules and rule-weights is difficult, and a natural question is how well the associated optimization problem can be solved, and how scalable such methods are. We use an easier-to-solve LP formulation.

**Hybrid methods.** In RNNLogic (Qu et al., 2021), the rules are generated using an RNN, and rule weights are later calculated via a probabilistic model. Such a separation of rule generation and weight calculation can be found in earlier work (e.g., Kok & Domingos (2005)), but in RNNLogic new rule generation is influenced by the calculated weights of previously generated rules. Our column generation method and AnyBURL have the same property.

**Reinforcement Learning.** Recent attempts to use reinforcement learning (RL) to search for rules can be found in MINERVA (Das et al., 2018), MultiHopKG (Lin et al., 2018), M-Walk (Shen et al., 2018) and DeepPath (Xiong et al., 2017). The first three papers use RL to explore relational paths conditioned on a specific query, and use RNNs to encode and construct a graph-walking agent.

**Rule types/Rule combinations.** AnyBURL generates rules corresponding to different types of paths (acyclic or cyclic, in their notation) which may be entity-dependent or independent. As mentioned before, NLIL goes beyond simple chain-like rules. Recently Teru et al. (2020) use subgraphs to perform reasoning, and not just paths. We generate *weighted, entity-independent, chain-like rules* as in NeuralLP. Our scoring function combines rule scores via a linear combination. For a rule $r$ and a pair of entities $a, b$, the rule score is just 1 if there exists a relational path from $a$ to $b$ following the rule $r$ and 0 otherwise. We use rule weights as a measure of importance but they are not probabilities. For Neural LP, DRUM, RNNLogic and other comparable codes, the scoring functions depend on the set of paths from $a$ to $b$ associated with $r$. Finally, AnyBURL uses maximum confidence scores rather than sums of confidence scores; further, its scoring function returns a vector of sorted scores, and two score vectors are compared lexicographically.

**Scalability/Compact Rule sets.** As noted above, many recent papers use RNNs in the process of finding chain-like rules and this can lead to expensive computation times. On the other hand, bottom-up rule-learners such as AnyBURL are much faster. The main focus of our work is obtaining compact rule sets for the sake of interpretability while maintaining scalability via LP models and column generation. We impose explicit constraints to maintain compactness, and these constraints influence new rule generation. The new rules must perform well with previously generated/selected rules. NeuralLP, for example, usually returns compact rule sets while AnyBURL returns a very large number of rules (and does not prune discovered rules for interpretability), and RNNLogic is somewhere in between (though the number of output rules can be controlled).

## 2.2 EMBEDDING-BASED METHODS

An alternative approach to KG reasoning is based on embedding entities and relations in the KG into (possibly different) vector spaces. For example, suppose one finds a vector $v_a \in \mathbb{R}^k$ for each node $a$ in the knowledge graph and a function $T_r : \mathbb{R}^k \to \mathbb{R}^k$ for each relation $r$ such that $T_r(v_a) \approx v_b$

whenever $(a, r, b)$ is a fact in the graph. Then, for a pair of entities $a$ and $b$, one could assert that $(a, r', b)$ is a fact (assuming it is not present in the graph) if $T_{r'}(v_a) \approx v_b$. Well known papers in this area are Sun et al. (2019), Bordes et al. (2013), Dettmers et al. (2018), Lacroix et al. (2018), Trouillon et al. (2016), Balažević et al. (2019), Nayyeri et al. (2021), and Chami et al. (2020).

There are a number of papers which combine embeddings and rules in different ways. In rule-injected embedding models such as RUGE (Guo et al., 2018), LogicENN (Nayyeri et al., 2019), and ComplEx-NNE-AER (Ding et al., 2018), the goal is to obtain embeddings that are consistent with prior rules (known before the training process). On the other hand, RNNLogic combines rules and embeddings to give more precise scores to candidate answers to queries of the form $(a, r, ?)$. In others (Lin et al., 2018), information from embeddings is used to obtain better rules.

Though embedding-based methods are better than most rule-based methods, AnyBURL is a fast rule-based method with comparable predictive performance to embedding based methods. However, AnyBURL resembles embedding-based methods in that the output KG representation has explicit entity dependence (the generated rules have constants which are often entities).

## 3 MODEL

We propose a linear programming model inspired by LP boosting methods for classification using classical *column generation* techniques (Demiriz et al., 2002; Golderg, 2012; Eckstein & Goldberg, 2012; Eckstein et al., 2019; Dash et al., 2018). Our goal is to create a weighted linear combination of first-order logic rules to be used as a scoring function for KG link prediction. In principle, our model has exponentially many variables corresponding to the possible rules, but our solution approach uses column generation to deal with this issue. We start with few initial candidate rules, find important rules and associated weights, and then generate additional rules that have the potential to improve the overall solution. Previously generated rules influence the generation of new rules (as in RNNLogic).

**Knowledge graphs:** Let $V$ be a set of entities, and let $\mathcal{R}$ be a set of $n$ binary relations defined over the domain $V$. A *knowledge graph* represents a collection of *facts* $\mathcal{F} \subseteq V \times \mathcal{R} \times V$ as a labeled, directed multigraph $\mathcal{G}$. Let $\mathcal{F} = \{(t^i, r^i, h^i) : i = 1, \ldots |\mathcal{F}|\}$ where $t^i \neq h^i \in V$, and $r^i \in \mathcal{R}$. The nodes of $\mathcal{G}$ correspond to entities in $\mathcal{F}$ and the edges to facts in $\mathcal{F}$: if $(t, r, h)$ is a fact in $\mathcal{F}$, then $\mathcal{G}$ has a directed edge $(t, h)$ labeled by the relation $r$, depicted as $t \xrightarrow{r} h$. Here $t$ is the *tail* of the directed edge, and $h$ is the *head*. We let $E$ stand for the list of directed edges in $\mathcal{G}$. For each fact $(t, r, h)$, we say that $r(h, t)$ is true. Practical KGs are assumed to be incomplete: missing facts that can be defined over $V$ and $\mathcal{R}$ are not assumed to be incorrect. The *knowledge graph link prediction* task consists of taking a knowledge graph as input, and then answering a list of queries of the form $(t, r, ?)$ and $(?, r, h)$, constructed from facts $(t, r, h)$ in a test set. The query $(t, r, ?)$ asks for a head entity $h$ such that $(t, r, h)$ is a fact, given a tail entity $t$ and a relation $r$. A collection of facts $\mathcal{F}$ is divided into a training set $\mathcal{F}_{tr}$, a validation set $\mathcal{F}_v$, and a test set $\mathcal{F}_{te}$, the KG $\mathcal{G}$ corresponding to $\mathcal{F}_{tr}$ is constructed and a scoring function (for link prediction) is learnt from $\mathcal{G}$ and evaluated on the test set.

**Goal:** For each relation $r$ in $\mathcal{G}$, find a set of *closed, chain-like* rules $R_1, \ldots, R_p$ and positive weights $w_1, \ldots w_p$ where each rule $R_i$ has the form

$$r_1(X, X_1) \wedge r_2(X_1, X_2) \wedge \cdots \wedge r_l(X_{l-1}, Y) \to r(X, Y). \tag{1}$$

Here $r_1, \ldots, r_l$ are relations from $\mathcal{R}$ represented in $\mathcal{G}$, and the *length* of the rule is $l$. The interpretation of this rule is that if for some entities (or nodes) $X, Y$ of $\mathcal{G}$ there exist entities $X_1, \ldots X_l$ of $\mathcal{G}$ such that $r_1(X, X_1), r_l(X_{l-1}, Y)$ and $r_j(X_{j-1}, X_j)$ are true for $j = 2, \ldots l - 1$, then $r(X, Y)$ is true. We refer to the conjunction of relations in (1) as the clause associated with the rule $R_i$. Thus each clause $C_i$ is a function from $V \times V$ to $\{0, 1\}$, and we define $|C_i|$ to be the number of relations in $C_i$. Clearly, $C_i(X, Y) = 1$ for entities $X, Y$ in $\mathcal{G}$ if and only if there is a *relational path* of the form

$$X \xrightarrow{r_1} X_1 \cdots X_{l-1} \xrightarrow{r_l} Y.$$

Our learned scoring function for relation $r$ is simply

$$f_r(X, Y) = \sum_{i=1}^{p} w_i C_i(X, Y) \text{ for all } X, Y \in V. \tag{2}$$

As discussed earlier, there are different ways to construct scoring functions from rules and rule weights. The linear nature of our scoring function is intimately linked to the use of LP to learn

the function. Given a query $(t, r, ?)$ constructed from a fact $(t, r, h)$ from the test set, we use the approach from Bordes et al. (2013) where the score $f_r(t, v)$ is calculated for every entity $v \in V$, and the *rank* of the correct entity $h$ is calculated from the scores of all entities in the *filtered* setting. We similarly calculate the rank of $t$ for the query $(?, r, h)$. We then compute standard metrics such as MRR (mean reciprocal rank), Hits@1, Hits@3, and Hits@10. An issue in rank computation is that multiple entities (say $e'$ and $e''$) can get the same score for a query. Different treatment of equal scores can lead to significantly different MRR values (Sun et al., 2020). See the Appendix. We use *random break* ranking (an option available in NeuralLP, where the correct entity is compared against all entities and any ties in scores are broken randomly.

**New LP model for rule learning for KG Link Prediction.** Let $\mathcal{K}$ denote the set of clauses of possible rules of the form (1) with maximum rule length $L$. Clearly, $|\mathcal{K}| = n^L$, where $n$ is the number of relations. Let $E_r$ be the set of edges in $\mathcal{G}$ labeled by relation $r$, and assume that $|E_r| = m$. Let the $i$th edge in $E_r$ be $(X_i, Y_i)$. We compute $a_{ik}$ as $a_{ik} = C_k(X_i, Y_i)$: $a_{ik}$ is 1 if and only if there is a relational path associated with the clause $C_k$ from $X_i$ to $Y_i$.

Furthermore, let $\text{neg}_k$ be a number associated with the number of "nonedges" $(X', Y')$ from $(V \times V) \setminus E_r$ for which $C_k(X', Y') = 1$. We calculate $\text{neg}_k$ for the $k$th rule as follows. We consider the tail node $t$ and head node $h$ for each edge in $E_r$. We compute the set of nodes $S$ that can be reached by a path induced by the $k$th rule starting at the tail. If there is no edge from $t$ to a node $v$ in $S$ labeled by $r$, we say that $v$ is an invalid end-point. Let $\text{right}_k$ be the set of such invalid points. We similarly calculate the set $\text{left}_k$ of invalid start-points based on paths ending at $h$ induced by the $k$th rule. The total number of invalid start and end points for all tail and head nodes associated with edges in $E_r$ is $\text{neg}_k = |\text{right}_k| + |\text{left}_k|$. For a query of the form $(t, r, ?)$ where $t$ is a tail node of an edge in $E_r$, the scoring function defined by the $k$th rule alone gives a positive and equal score to all nodes in $\text{right}_k$.

Our model for rule-learning is given below.

(LPR)

$$z_{min} = \quad \min \quad \sum_{i=1}^{m} \eta_i + \tau \sum_{k \in \mathcal{K}} \text{neg}_k w_k \tag{3}$$

$$s.t. \quad \sum_{k \in \mathcal{K}} a_{ik} w_k + \eta_i \geq 1 \text{ for all } i \in E_r \tag{4}$$

$$\sum_{k \in \mathcal{K}} (1 + |C_k|) w_k \leq \kappa \tag{5}$$

$$w_k \in [0, 1] \text{ for all } k \in \mathcal{K}. \tag{6}$$

The continuous variable $w_k$ is restricted to lie in $[0, 1]$ and is positive if and only if clause $k \in \mathcal{K}$ is a part of the scoring function (2). The parameter $\kappa$ is an upper bound on the *complexity* of the scoring function (defined as the number of clauses plus the number of relations across all clauses). The variable $\eta_i$ is a penalty variable which is positive if the scoring function defined by positive $w_k$s gives a value less than 1 to the $i$th edge in $E_r$. Therefore, the $\sum_{i=1}^{m} \eta_i$ portion of the objective function attempts to maximize $\sum_{i=1}^{m} \min\{f_r(X_i, Y_i), 1\}$, i.e., it attempts to approximately maximize the number of facts in $E_r$ that are given a "high-score" of 1 by $f_r$. In addition, we have the parameter $\tau > 0$ which represents a tradeoff between how well our weighted combination of rules performs on the known facts (gives positive scores), and how poorly it performs on some negative samples or "unknown" facts. We make this precise shortly.

Maximizing the MRR is a standard objective for KG link prediction problems and thus the objective function of LPR is only an approximation; see the next Theorem (the proof is given in the Appendix). In spite of this fact, we can obtain state-of-the-art prediction rules using LPR.

**Theorem 1.** *Let IPR be the integer programming problem created from LPR by replacing equation (6) by $w_k \in \{0, 1\}$ for all $k \in \mathcal{K}$, and letting $\tau = 0$. Given an optimal solution with objective function value $\gamma$, one can construct a scoring function such that $1 - \gamma/m$ is a lower bound on the MRR of the scoring function calculated by the optimistic ranking method, when applied to the training set triples.*

Assuming that the training set facts have a similar distribution to the test set facts, the theorem above justifies choosing IPR as an optimization formulation to find a high-quality collection of rules for a relation, assuming MRR calculation via optimistic ranking.

However, we use random break ranking in this paper. It is essential to perform negative sampling and penalize rules that create paths when there are no edges in order to produce good quality results. This is why we use $\tau > 0$ in LPR. We will now give an interpretation of $\sum_k \text{neg}_k w_k$. To compute the MRR of the scoring function $f_r$ in (2) applied to the training set, for each edge $(t, r, h) \in E_r$ we need to compute the rank of the answer $h$ to the query $(t, r, ?)$ – by comparing $f_r(t, r, v)$ with $f_r(t, r, h)$ for all nodes $v$ in $\mathcal{G}$ – and the rank of answer $t$ to the query $(?, r, h)$ – by comparing $f_r(v, r, h)$ with $f_r(t, r, h)$ for all nodes $v$. But $\sum_k \text{neg}_k w_k$ is exactly the sum of scores given by $f_r$ to all nodes in $\text{right}_k$ and $\text{left}_k$ and therefore we have the following proposition.

**Proposition 2.** *Let $(t, r, h)$ be an edge in $E_r$, and let $U(?, r, h)$ be the set of invalid answers for $(?, r, h)$ and let $U(t, r, ?)$ be the set of invalid answers to $(t, r, ?)$ in the filtered setting. Then*

$$\sum_{(t,r,h)\in E_r} \left( \sum_{v\in U(?,r,h)} f_r(v,r,h) + \sum_{v\in U(t,r,?)} f_r(t,r,v) \right) = \sum_{k\in\mathcal{K}} \text{neg}_k w_k.$$

In other words, rather than keeping individual scores of the form $f_r(v, r, h)$ and $f_r(t, r, v)$ small, we minimize the sum of these scores in LPR.

It is impractical to solve LPR given the exponentially many variables $w_k$, except when $n$ and $L$ are both small. For WN18RR (Dettmers et al., 2018), $n$ is only 22 (WN18RR has 11 relations, but we introduce a reverse relation $r^{-1}$ for each $r \in \mathcal{R}$, and create rules that include reverse relations) and thus setting $l$ to 3 does not lead to too many variables. An effective way to solve such large LPs is to use column generation where only a small subset of all possible $w_k$ variables is generated explicitly and the optimality of the LP is guaranteed by iteratively solving a *pricing* problem. We do not attempt to solve LPR to optimality. We start with an initial set of candidate rules $\mathcal{K}_0 \subset \mathcal{K}$ (and implicitly set all rule variables from $\mathcal{K} \setminus \mathcal{K}_0$ to 0). Let $\text{LPR}_0$ be the associated LP. We solve $\text{LPR}_0$, and then dynamically augment the set of candidate rules to create sets $\mathcal{K}_i$ such that $\mathcal{K}_0 \subset \mathcal{K}_1 \subset \cdots \subset \mathcal{K}$. If $\text{LPR}_i$ is the LP associated with $\mathcal{K}_i$ with optimal solution value $z^i_{min}$, then it is clear that a solution of $\text{LPR}_i$ is a solution of $\text{LPR}_{i+1}$, and therefore $z^{i+1}_{min} \leq z^i_{min}$. We attempt to have $z^{i+1}_{min} < z^i_{min}$ by taking the dual solution associated with an optimal solution of $\text{LP}_i$, and then trying to find a *negative reduced cost* rule, which we discuss shortly.

**Setting up the initial LP.** To set up $\mathcal{K}_0$ and the associated $\text{LP}_0$, we develop two heuristics. In Rule Heuristic 1, we generate rules of lengths one and two. For length one rules, we create a one-relation rule from a relation in $\mathcal{R} \setminus \{r\}$ if it labels a large number of edges from tail nodes to head nodes of edges in $E_r$. Similarly, to create rules of length two, we take each edge $(X, Y)$ in $E_r$ and select the relations from edges $(X, Z)$ in $E \setminus E_r$ and $(Z, Y)$ in $E$ to create a rule, again taking into account how frequently a length two rule creates paths from the tail nodes to head nodes of edges in $E_r$.

In Rule Heuristic 2, we take each edge $(X, Y)$ in $E_r$ and find a shortest path from $X$ to $Y$ contained in the edge set $E \setminus \{(X, Y)\}$ where the path length is bounded by a pre-determined maximum length. We then use the sequence of relations associated with the shortest path to generate a rule. We also use a path of length at least one more than the shortest path. Rules generated by any method (NeuralLP, DRUM, etc) can be used to set up $\mathcal{K}_0$.

**Adding new rules.** Each set $\mathcal{K}_i$ for $i > 0$, is constructed by adding new rules to the set $\mathcal{K}_{i-1}$. We use a modified version of Heuristic 2 to generate the additional rules. In this version of the heuristic, we use the dual variable values associated with the optimal solution of $\text{LPR}_{i-1}$. Let $\delta_i \geq 0$ for all $i \in E_r$ be dual variables corresponding to constraints (4). Let $\lambda \leq 0$ be the dual variable associated with the constraint (5). Given a variable $w_k$ which is zero in a solution of $\text{LPR}_i$ and associated dual solution values $\bar{\delta}$ and $\bar{\lambda}$, the reduced cost $\text{red}_k$ for this variable is given by

$$\text{red}_k = \tau \, \text{neg}_k - \sum_{i\in E_r} a_{ik} \bar{\delta}_i - (1 + |C_k|)\bar{\lambda}$$

If $\text{red}_k < 0$, then increasing $w_k$ from zero may reduce the LP solution value.

To try make the reduced cost negative, we sort the dual values $\bar{\delta}_j$ in decreasing order, and then go through the associated indices $j$, and create rules $k$ such that $a_{jk} = 1$ via a shortest path calculation. That is, we take the corresponding edge $(X, Y)$ in $E_r$, find the shortest path between $X$ and $Y$ and generate a new rule with the sequence of relations in that path. We limit the number of rules generated so that $\mathcal{K}_i$ is only slightly larger than $\mathcal{K}_{i-1}$. More precisely, we set $|\mathcal{K}_i| - |\mathcal{K}_{i-1}| \leq 10$. We use the

| Algorithm | Kinship | | | | UMLS | | | |
|---|---|---|---|---|---|---|---|---|
| | MRR | H@1 | H@3 | H@10 | MRR | H@1 | H@3 | H@10 |
| ComplEx-N3 | 0.889 | 0.824 | 0.950 | 0.986 | 0.962 | 0.934 | 0.990 | 0.996 |
| TuckER | 0.891 | 0.826 | 0.950 | 0.987 | 0.914 | 0.837 | 0.991 | 0.997 |
| † ConvE | 0.83 | 0.74 | 0.92 | 0.98 | 0.94 | 0.92 | 0.96 | 0.99 |
| AnyBURL | 0.653 | 0.523 | | 0.924 | 0.952 | 0.931 | | 0.990 |
| NeuralLP | 0.652 | 0.520 | 0.726 | 0.925 | 0.750 | 0.601 | 0.876 | 0.958 |
| DRUM | 0.566 | 0.404 | 0.663 | 0.910 | 0.845 | 0.722 | 0.959 | 0.991 |
| RNNLogic | *0.687 | 0.566 | 0.756 | 0.929 | *0.748 | 0.618 | 0.849 | 0.928 |
| LPRules | 0.746 | 0.639 | 0.816 | 0.959 | 0.869 | 0.812 | 0.917 | 0.970 |

Table 1: Comparison of results on Kinship and UMLS. The results from NeuralLP, DRUM, and our code use the random break metric. *We modified the RNNLogic code to compute the MRR values based on the random break metric and the values obtained were the same as the values of the original MRR computation up to three decimal places. † ConvE results are from the original paper.

dual values to indicate which facts are not currently implied by the existing set of chosen rules. If the reduced cost of a new rule is nonnegative, then we do not add that rule to $\mathcal{K}_{i-1}$.

## 4 EXPERIMENTS

We conduct experiments on knowledge graph completion tasks with six datasets: Kinship (Denham, 1973), UMLS (McCray, 2003), FB15k-237 (Toutanova & Chen, 2015), WN18RR (Dettmers et al., 2018), YAGO3-10 (Mahdisoltani et al., 2015) and DB111K-174 (Hao et al., 2019). In Table 5, we give properties of the datasets: the number of entities, relations, and the number of facts in the training, testing and validation data sets. The partition of FB15k-237, WN18RR, and YAGO3-10 into training, testing, and validation data sets is standard. We chose the partition for UMLS and Kinship used in Dettmers et al. (2018) and the partition for DB111K-174 given in Cui et al. (2021).

### 4.1 EXPERIMENTAL SETUP

We denote the reverse relation for each relation $r \in \mathcal{R}$ by $r^{-1}$. For each fact $(t, r, h)$ in the training set, we implicitly introduce the fact $(h, r^{-1}, t)$ doubling the number of relations and facts. For each *original* relation $r$ in the training set, we create a scoring function of the form $f_r(X, Y)$ in (2). For each test set fact $(t, r, h)$ we create two queries $(t, r, ?)$ and $(?, r, h)$, and use $f_r$ to predict answers to these queries. For each entity $e$ in $\mathcal{G}$, we calculate the scores $f_r(t, e)$ and $f_r(e, h)$ – here $e$ is treated as a candidate solution to the queries $(t, r, ?)$ and $(?, r, h)$ – and then use the filtered ranking method in Bordes et al. (2013) to calculate a ranking for the correct answer (namely $(t, r, h)$) to the above queries. Ranks are computed using the random break method (option in NeuralLP (Yang et al., 2017)), and these are used to compute MRR and Hits@$k$ (for $k = 1, 3, 10$) across all test facts.

We compare our results with the published embedding-based methods ConvE (Dettmers et al., 2018), ComplEx-N3 (Lacroix et al., 2018), TuckER (Balažević et al., 2019), RotatE (Sun et al., 2019), 5*E (Nayyeri et al., 2021), and ATTH (Chami et al., 2020). We obtained results for ComplEx-N3 and TuckER by running on our machines using the best published hyperparameters (if available). The results for the other embedding-based methods were taken from either the original papers or from Dettmers et al. (2018) or Qu et al. (2021). These codes do not implement random break ranking for equal scores, and some instead use nondeterministic ranking (Berrendorf et al., 2021), i.e., they sort entity scores before ranking. We compare with the rule-based methods NeuralLP, DRUM, and RNNLogic. We also run AnyBURL for 100 seconds. Additional comparisons can be found in the Appendix. We obtain results for NeuralLP and DRUM using default parameters and random break score ranking. We modify the RNNLogic code to implement random break ranking, and use the

defaults suggested for different datasets. RNNLogic claims to use midpoint ranking, but actually implements a harmonic mean of possible reciprocal ranks in the presence of equal scores while calculating the MRR, which yields a very slightly larger number than midpoint ranking based MRR.

We ran two variants of our code which we call "LPRules". In the first variant, we create $LPR_0$ by generating rules using Rule Heuristic 1 and Rule Heuristic 2, and then solve $LPR_0$ to obtain rules. As the results are satisfactory for smaller datasets, we do not perform column generation. In the second variant, which we use only for the largest instances, we create $LPR_0$ with an empty set of rules, and then perform 5 iterations consisting of generating up to 10 rules using the modified version of Rule Heuristic 2 followed by solving the new LP. In other words, we create and solve $LPR_i$ for $i = 0, \ldots 5$.

We search for the best values of $\tau$ and $\kappa$ for each relation. We dynamically let $\bar{\kappa}$ equal the length of the longest rule generated plus one. We then perform 20 iterations where, at the $i$th iteration, we set $\kappa$ to $i\bar{\kappa}$. For each combination of $\tau$ and $\kappa$, we take the optimal weighted combination of rules and compute the MRR on the validation data set, and select those $\tau$ and $\kappa$ that yield the best MRR. We set the maximum rule length to 6 for WN18RR, and 3 for YAGO3-10, and 4 for the other datasets. Thus $\kappa \leq 100$ except for WN18RR. We search for the best $\tau$ from the list of values in Table 11.

| | FB15k-237 | | | | WN18RR | | | |
|---|---|---|---|---|---|---|---|---|
| Algorithm | MRR | H@1 | H@3 | H@10 | MRR | H@1 | H@3 | H@10 |
| ComplEx-N3 | 0.362 | 0.259 | 0.397 | 0.555 | 0.469 | 0.434 | 0.481 | 0.545 |
| TuckER | 0.353 | 0.259 | 0.390 | 0.538 | 0.464 | 0.436 | 0.477 | 0.517 |
| † ConvE | 0.325 | 0.237 | 0.356 | 0.501 | 0.43 | 0.40 | 0.44 | 0.52 |
| † RotatE | 0.338 | 0.241 | 0.375 | 0.533 | 0.476 | 0.428 | 0.492 | 0.571 |
| ‡ 5*E | 0.37 | 0.28 | 0.40 | 0.56 | 0.50 | 0.45 | 0.51 | 0.59 |
| AnyBURL | 0.278 | 0.212 | | 0.444 | 0.479 | 0.448 | | 0.555 |
| NeuralLP | 0.222 | 0.160 | 0.240 | 0.349 | 0.381 | 0.367 | 0.386 | 0.409 |
| DRUM | 0.225 | 0.160 | 0.245 | 0.355 | 0.381 | 0.367 | 0.389 | 0.410 |
| RNNLogic | ♯0.288 | 0.208 | 0.315 | 0.445 | *0.451 | 0.415 | 0.474 | 0.524 |
| LPRules | 0.255 | 0.170 | 0.270 | 0.402 | 0.459 | 0.422 | 0.477 | 0.532 |

Table 2: Comparison of results on FB15k-237 and WN18RR. *The RNNLogic MRR value for WN18RR is obtained via random break ranking. ♯ We could not run RNNLogic on FB15k-237, and report numbers from the original paper. † The results for ConvE and RotatE were taken from Qu et al. (2021). ‡ The results for 5*E were taken from Nayyeri et al. (2021).

## 4.2 RESULTS

In Tables 1-3, we give values for different metrics obtained with the listed codes, first for embedding methods, then for AnyBURL, then for rule-based methods (if available), and then for our code. We place AnyBURL in a separate category as it is rule-based, but is similar to embedding-based methods in its entity-dependent KG representation. Henceforth, *rule-based codes* does not refer to AnyBURL. All our experiments (with rule-based codes) are performed on a machine with 128 GBytes of memory, and four 2.8 Intel Xeon E7-4890 v2 processors, each with 15 cores, for a total of 60 cores. We use coarse-grained parallelism in our code, and execute rule generation for each relation on a different thread, and solve LPs with IBM CPLEX (IBM, 2019). If one relation dominates the others (w.r.t number of facts, as in YAGO3-10), then our implementation becames essentially single-threaded.

In Table 1, we present results on Kinship and UMLS. Our method obtains better results than NeuralLP, DRUM, and RNNLogic on Kinship across all measures, and better MRR and Hits@1 values than these three codes on UMLS. This is true even though we generate relatively compact rules (see Table 4 ) and also use very simple rule generation heuristics. Therefore, for these datasets, even trivial rule generation heuristics suffice when used in conjunction with a nontrivial weight generation algorithm. The embedding methods are much better across all metrics. In Table 2, we present

results on FB15k-237 and WN18RR. RNNLogic did not successfully terminate for FB15k-237 on our machine, and we take the published result. Our results for FB15k-237 are better than NeuralLP and DRUM but worse than RNNLogic. The best values for embedding-based methods are much better than for all rule-based methods. For WN18RR, our results are better than the other rule-based methods, even while taking significantly less computing time, see Table 6 in the Appendix. This better scaling behaviour allows us to tackle large datasets such as YAGO3-10, which we give results for in Table 3. We use column generation, generating 10 columns in each iteration, and iterate only 5 times for a total of 50 candidate rules per relation. To compute $\text{neg}_k$, we sample 20% of the edges from $E_r$, and compute the number of invalid paths that start at the tails of these edges, and end at the heads of these edges. For YAGO3-10 and DB111K-174, our column generation approach becomes essential. We are simply unable to process a very large number of rules. The ability to generate a small number of rules, and then use the dual values to focus on "uncovered" facts (not implied by previous rules) and generate new rules covering these uncovered facts is very helpful.

| | YAGO3-10 | | | | DB111K-174 | | | |
|---|---|---|---|---|---|---|---|---|
| Algorithm | MRR | H@1 | H@3 | H@10 | MRR | H@1 | H@3 | H@10 |
| ComplEx-N3 | 0.574 | 0.499 | 0.619 | 0.705 | 0.421 | 0.344 | 0.459 | 0.563 |
| TuckER | 0.265 | 0.184 | 0.290 | 0.426 | 0.345 | 0.247 | 0.397 | 0.529 |
| † ConvE | 0.44 | 0.35 | 0.49 | 0.62 | | | | |
| ‡ RotatE | 0.495 | 0.402 | 0.550 | 0.670 | | | | |
| ♯ ATTH | 0.568 | 0.493 | 0.612 | 0.702 | | | | |
| AnyBURL | 0.543 | 0.486 | | 0.659 | 0.391 | 0.339 | | 0.512 |
| LPRules | 0.449 | 0.367 | 0.501 | 0.684 | 0.363 | 0.312 | 0.390 | 0.460 |

Table 3: Comparison of results on YAGO3-10 and DB111K-174 using random break metric. † The results for ConvE are taken from Dettmers et al. (2018). ‡ The results for RotatE are from Sun et al. (2019). ♯ The results for ATTH are from Chami et al. (2020).

In Table 4, we confirm that we obtain compact rule sets as measured by the average number of rules in the final solution, given in column '#rules'. These final rules are selected from a few hundred to a thousand generated rules, other than in YAGO3-10, where we only generate 50 rules. We get better MRR for similar levels of sparsity compared to NeuralLP and RNNLogic. For WN18RR and UMLS, we obtain state-of-the-art results with few rules. We could not extract rules from DRUM. RNNLogic chooses top-$K$ rules for testing ($K$ is an input parameter), and we run with the default $K = 200$ and $K = 20$. AnyBURL generates many rules and does not prune previously generated rules.

| | Kinship | | UMLS | | FB15k-237 | | WN18RR | | YAGO3-10 | |
|---|---|---|---|---|---|---|---|---|---|---|
| Algorithm | MRR | #rules | MRR | #rules | MRR | #rules | MRR | #rules | MRR | #rules |
| AnyBURL | 0.653 | 13334 | 0.952 | 6228 | 0.278 | 1360 | 0.479 | 4783 | 0.543 | 12852 |
| NeuralLP | 0.652 | 10.2 | 0.750 | 14.2 | 0.222 | 8.3 | 0.381 | 14.6 | | |
| RNNLogic-20 | 0.600 | 20 | 0.706 | 20 | | | 0.416 | 20 | | |
| RNNLogic-200 | 0.687 | 200 | 0.677 | 200 | | | 0.451 | 200 | | |
| LPRules | 0.746 | 21.0 | 0.848 | 4.2 | 0.255 | 14.2 | 0.459 | 15.6 | 0.449 | 7.8 |

Table 4: MRR and average number of rules selected per relation

## 5 CONCLUSION

Existing methods to obtain logic rules for knowledge graph completion can be fairly time consuming. Our relatively simple linear programming formulation for selecting weighted logical rules and associated solution algorithm can return state-of-the-art results for a number of standard KG datasets even with sparse collections of rules, and much faster than existing methods.

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

## A    APPENDIX

### A.1    DATASET DETAILS AND RUNNING TIMES

| Datasets | # Relations | # Entities | # Train | # Test | # Valid |
|---|---|---|---|---|---|
| Kinship | 25 | 104 | 8544 | 1074 | 1068 |
| UMLS | 46 | 135 | 5216 | 661 | 652 |
| WN18RR | 11 | 40943 | 86835 | 3134 | 3034 |
| FB15k-237 | 237 | 14541 | 272115 | 20466 | 17535 |
| YAGO3-10 | 37 | 123182 | 1079040 | 5000 | 5000 |
| DB111K-174 | 298 | 98336 | 474123 | 65851 | 118531 |

Table 5: Sizes of datasets.

In Table 6, we give run times (in minutes) on the different datasets. The results in the top section of the table were obtained on the 60 core machine described above and the results in the bottom section of the table were obtained on a machine with 16 CPUs and 1 GPU. Our times include the time to evaluate the solution on the test set. AnyBURL is run for a fixed time of 100 seconds. For WN18RR, DRUM and NeuralLP take 400 minutes or more, and RNNLogic takes over 100 minutes, whereas we take 11 minutes and less than 2 minutes with LPRules. We note that we use maximum rule length of 6 for WN18RR, but we do not know the maximum path lengths used in the other codes. DRUM takes over 8 days for FB15K-237 (as does NeuralLP on a different machine), which is why we do not run these codes on YAGO3-10 and DB111K-174. Our code can be sped up further if fine-grained parallelism were used (see the Appendix A.5).

### A.2    ADDITIONAL COMPARISONS

It is challenging to compare published codes for KGC as they differ in important ways.

|  | Kinship | UMLS | WN18RR | FB15k-237 | YAGO3-10 | DB111K-174 |
|---|---|---|---|---|---|---|
| AnyBURL | 1.7 | 1.7 | 1.7 | 1.7 | 1.7 | 1.7 |
| NeuralLP | 1.6 | 1.1 | 399.9 | | | |
| DRUM | 3.2 | 2.8 | 505.9 | 12053.3 | | |
| RNNLogic | 108.8 | 133.4 | 104.0 | | | |
| LPRules | 0.5 | 0.2 | 11.0 | 234.5 | 1648.4 | 152.4 |
| ComplEx-N3 | 2.6 | 1.7 | 195.9 | 238.6 | 2024.8 | 745.2 |
| TuckER | 8.9 | 5.5 | 266.6 | 407.5 | 2894.7 | 6161.1 |
| LPRules | 2.7 | 0.4 | 11.7 | 267.3 | 1245.9 | 131.0 |

Table 6: Wall clock run times in minutes when running in parallel on a 60 core machine for the top group of results and a machine with 16 CPUs and 1 GPU for the bottom results.

a. Entity Ranking: There are different ways (Berrendorf et al., 2021; Sun et al., 2020) of ranking entities $e$ that form candidate solutions to queries $(t, r, ?)$ and $(?, r, h)$ constructed from a test set fact $(t, r, h)$, and the choice of the ranking method may have a significant effect on the final MRR, especially for rule-based methods, see Table 7. Assume $m$ entities receive a strictly better score than the true answer $h$ for the query $(t, r, ?)$ and $n$ entities receive the same score. Some ranking methods listed in Berrendorf et al. (2021) are: $h$ receives a rank of $m + 1$ (*optimistic*), i.e., the best among all equally-scored entities, or $m + n$ (*pessimistic*), or $k$ where $k$ is a random number between $m + 1$ and $m + n$ (*random* (Sun et al., 2020)). It is also observed in Berrendorf et al. (2021) that numerous codes give a *nondeterministic* rank between $m + 1$ and $m + n$ based on the position of the correct entity in a sorted order of equal score entities. Other options are a rank of $m + (n - m)/2$ (*midpoint*) (Qu et al., 2021) or *random break*, where the correct entity is compared against all entities and any ties in scores are broken randomly. We observe that *random* and *random break* are not the same: the probability of getting any rank between $m + 1$ and $m + n$ is the same in *random*, whereas $m + (n - m)/2$ is more probable as a rank than $m + 1$ or $m + n$ in random break. The choice of the ranking method also depends on whether one assumes a closed or open world hypothesis. The NTP and NTP-$\lambda$ codes referred to in Table 8 use an *optimistic* ranking method, whereas MINERVA uses a *nondeterministic* ranking method.

b. Query construction: Some codes, such as MINERVA, only remove right entities from a fact $(t, r, h)$ in the test set to construct the query $(t, r, ?)$, and do not evaluate performance on the query $(?, r, h)$.

c. Hyperparameters and Experimental Setup: It was shown recently (Ruffinelli et al., 2020) that some older, seemingly lower-quality codes could be made to perform better than more recent codes with appropriate choices of hyperparameters.

Random ranking is proposed in Sun et al. (2020) as a suitable method for KGC, but the codes NeuralLP and DRUM provide random-break ranking (which is closer to midpoint ranking) instead of random ranking. We earlier compared our code with these codes and with RNNLogic using the random-break ranking method, and two-sided query construction during testing. In the next table, we show that our code returns very different numbers if we use optimistic ranking, but similar results if we use midpoint ranking.

In Table 8, we give a comparison with results published in the MINERVA paper (Das et al., 2018) that are obtained with right-entity removal only, in query construction. We modify our code to construct queries in a similar fashion, though we still use random-break ranking, The grouping of results by embedding based methods, rule based methods, and those obtained by our code is as before. MINERVA and ConvE use nondeterministic (Berrendorf et al., 2021) ranking, as they sort scores before ranking, but NTP does not. The NTP code refers to the ranking method in ComplEx and says that "we calculate the rank from only those corrupted triples that have a higher score" - i.e., they use optimistic ranking. In our opinion, this accounts for the unusually high scores for UMLS obtained with NTP-$\lambda$. When we use optimistic ranking for UMLS, we obtain an **MRR of 0.967**. ComplEx

|  | WN18RR | | | |
| Metric | MRR | H@1 | H@3 | H@10 |
|---|---|---|---|---|
| Optimistic | 0.658 | 0.603 | 0.678 | 0.768 |
| Midpoint | 0.455 | 0.415 | 0.474 | 0.532 |
| Random Break | 0.459 | 0.422 | 0.477 | 0.532 |

Table 7: Results on WN18RR obtained with different ways of dealing with equal scores.

also uses optimistic ranking. Though we cannot prove it, we suspect that some embedding based methods (especially, ConvE, RotatE, and TuckER; see Sun et al. (2020)) do not return much better MRR values when using optimistic ranking instead of random-break ranking. This can happen if few entities get equal scores. We cannot locate the ranking method used in DistMult.

|  | Kinship | | | | UMLS | | | |
| Algorithm | MRR | H@1 | H@3 | H@10 | MRR | H@1 | H@3 | H@10 |
|---|---|---|---|---|---|---|---|---|
| ComplEx | 0.838 | 0.754 | 0.910 | 0.980 | 0.894 | 0.823 | 0.962 | 0.995 |
| ConvE | 0.797 | 0.697 | 0.886 | 0.974 | 0.933 | 0.894 | 0.964 | 0.992 |
| DistMult | 0.878 | 0.808 | 0.942 | 0.979 | 0.944 | 0.916 | 0.967 | 0.992 |
| † NTP | 0.612 | 0.500 | 0.700 | 0.777 | 0.872 | 0.817 | 0.906 | 0.970 |
| † NTP-$\lambda$ | 0.793 | 0.759 | 0.798 | 0.878 | 0.912 | 0.843 | 0.983 | 1.000 |
| NeuralLP | 0.619 | 0.475 | 0.707 | 0.912 | 0.778 | 0.643 | 0.869 | 0.962 |
| MINERVA | 0.720 | 0.605 | 0.812 | 0.924 | 0.825 | 0.728 | 0.900 | 0.968 |
| LPRules | 0.776 | 0.682 | 0.836 | 0.966 | 0.887 | 0.841 | 0.924 | 0.971 |

Table 8: Comparison with results on Kinship and UMLS taken from the MINERVA paper. Queries are constructed from test set facts by right entity removal. Our code uses random break ranking. † NTP uses optimistic ranking.

|  | FB15k-237 | | | | WN18RR | | | |
| Algorithm | MRR | H@1 | H@3 | H@10 | MRR | H@1 | H@3 | H@10 |
|---|---|---|---|---|---|---|---|---|
| ComplEx | 0.394 | 0.303 | 0.434 | 0.572 | 0.415 | 0.382 | 0.433 | 0.480 |
| ConvE | 0.410 | 0.313 | 0.457 | 0.600 | 0.438 | 0.403 | 0.452 | 0.519 |
| DistMult | 0.370 | 0.275 | 0.417 | 0.568 | 0.433 | 0.410 | 0.441 | 0.475 |
| NeuralLP | 0.227 | 0.166 | 0.248 | 0.348 | 0.463 | 0.376 | 0.468 | 0.657 |
| Path-Baseline | 0.227 | 0.169 | 0.248 | 0.357 | 0.027 | 0.017 | 0.025 | 0.046 |
| MINERVA | 0.293 | 0.217 | 0.329 | 0.456 | 0.448 | 0.413 | 0.456 | 0.513 |
| † M-Walk | 0.232 | 0.165 | 0.243 | | 0.437 | 0.414 | 0.445 | |
| LPRules | 0.350 | 0.261 | 0.383 | 0.533 | 0.486 | 0.443 | 0.511 | 0.571 |

Table 9: Comparison with results on FB15k-237 and WN18RR taken from the MINERVA paper. Our code uses random break ranking. † The M-Walk results are taken from the associated paper.

In Table 9, we copy results for FB15K-237 and WN18RR – obtained by right-entity removal in query construction – from the MINERVA paper. We also copy results obtained with M-Walk (taken from the corresponding paper), as it uses the same query construction approach. We do not know the score ranking method used in M-Walk or in the NeuralLP runs from the MINERVA paper. We note that our code obtains the best value of MRR for WN18RR, and the best MRR value among all rule-based methods for FB15K-237. Thus, for the four datasets mentioned in this table and the previous one, our code returns state-of-the-art results among the rule-based methods compared here.

## A.3 PROOFS

We next present the proof of Theorem 1.

*Proof.* Let $(\bar{\eta}, \bar{w})$ be an optimal solution of IPR. By definition, $\bar{w}$ has $|\mathcal{K}|$ components, and $\bar{\eta}$ has $m = |E_r|$ components, all of which are binary, because of the form of the objective function. Let $C_k$ be the clause associated with rule $k$. By definition, we have $a_{ik} = C_k(X_i, Y_i)$. Consider the function

$$f(X, Y) = \vee_{k:\bar{w}_k=1} C_k(X, Y) = \vee_{k \in \mathcal{K}} \bar{w}_k C_k(X, Y). \tag{7}$$

Therefore, $f : V \times V \to \{0, 1\}$. As $(\bar{\eta}, \bar{w})$ satisfies equation (4), we have

$$\sum_{k \in \mathcal{K}} a_{ik} \bar{w}_k + \bar{\eta}_i \geq 1 \text{ for all } i \in E_r.$$

We can see that $f(X_i, Y_i) = 1$ if and only if $\sum_{k \in \mathcal{K}} a_{ik} \bar{w}_k \geq 1$, and therefore

$$f(X_i, Y_i) + \bar{\eta}_i \geq 1 \text{ for all } i \in E_r.$$

Therefore, either $f(X_i, Y_i) = 1$ or $\bar{\eta}_i = 1$. Furthermore, $f(X, Y)$ is a function for which fewest number of values $\bar{\eta}_i$ are 1 or the highest number of values $\bar{\eta}_i$ are 0, as $(\bar{\eta}, \bar{w})$ form an optimal solution of IPR. In other words, $f(X, Y)$ "covers" the largest number of edges of $E_r$ (covering means $f(X_i, Y_i) = 1$) among all possible functions that can be formed as a disjunction of rule clauses with complexity at most $\kappa$. For each $i$ such that $\bar{\eta}_i = 0$, we have

$$f(X_i, Y_i) = 1 \text{ and } f(X_i, Z) \leq 1 \text{ and } f(Z, Y_i) \leq 1$$

for all $Z \in V$ as $f(X, Y) = 0$ or 1 for any entities $X, Y \in V$. If we take the facts in the training set as a test set, and use $f(X, Y)$ as a scoring function and use *optimistic* ranking of scores, then for each $i$ such that $f(X_i, Y_i) = 1 \geq f(X_i, Z)$ and $f(X_i, Y_i) \geq f(Z, Y_i)$ for all entities $Z$, and therefore the rank of $Y_i$ is 1 among all entities $Z$ while scoring $(X_i, Z)$ (denoted by $rr_i$), and the rank of $X_i$ is 1 among all the entities $Z$ while scoring $(Z, Y_i)$ (denoted by $lr_i$). On the other hand $rr_i \geq 1$ and $lr_i \geq 1$ if $f(X_i, Y_i) = 0$. Therefore $1/rr_i \geq 1 - \bar{\eta}_i$ and $1/lr_i \geq 1 - \bar{\eta}_i$. The MRR of the prediction function is

$$(\sum_{i=1}^{m} \tfrac{1}{rr_i} + \sum_{i=1}^{m} \tfrac{1}{lr_i})/2m \quad \geq 2(m - \sum_{i=1}^{m} \bar{\eta}_i)/2m$$

$$= 1 - \frac{1}{m} \sum_{i=1}^{m} \bar{\eta}_i. \tag{8}$$

But $\sum_{i=1}^{m} \bar{\eta}_i$ is the optimal objective function value. Thus a lower value of $\sum_{i=1}^{m} \eta_i$ yields a higher lower bound on the MRR computed via optimistic ranking. $\square$

## A.4 ANALYSIS OF YAGO3-10

We obtained an MRR of 0.449 with LPRules. We next analyze our performance and observe that it is primarily due to the perfomance on two relations, namely *IsAffiliatedTo* and *playsFor*, which together account for 64.4% of the facts in the training set, and 63.3% of the facts in the test set. In Table 10, we provide the rules generated by LPRules for these two relations. Here *R_isAffiliatedTo* is the reverse of the relation *isAffiliatedTo* (and denoted by *isAffiliatedTo*$^{-1}$ in the main document). The rule and weight columns together give the weighted combination of rules generated for a relation. The MRR column gives the MRR value that would be generated if the test set consisted only of facts associated with the relation in the "Relation" column, and the weighted combination of rules

consisted only of the rules in the same line or above. Thus we can see that if we took only the first two rules for *isAffiliatedTo*, and the rules for *playsFor*, then the MRR would be at least 0.56 on the test set facts associated with these two relations. As these two relations account for 63.3% of the test set, just the four rules mentioned above would yield an MRR $\geq 0.56 \times 0.633 \approx 0.354$, as opposed to the MRR of 0.449 that we obtained.

The information in the table suggests a direct correlation between the relations *isAffiliatedTo* and *playsFor*. Indeed, we verify that for about 75% of the facts *(x, isAffiliatedTo, y)*, we also have *(x, playsFor, y)* as a fact in the training set. Similarly, 87% of the *playsFor* facts are explained by the isAffiliatedTo relation in the training set.

A natural question is the following: Is the second rule for *isAffiliatedTo* a "degenerate" rule and does it simply reduce to *isAffiliatedTo* because the entity *b* is the same as entity *a* when we traverse a relational path from *x* to *y* in the training KG. To give an example that this is not the case, consider the following fact in the test set: *(Pablo_Bonells, isAffiliatedTo, Club_Celaya)*. In the training data, the following three facts imply the previous fact by application of the second rule: *(Pablo_Bonells, isAffiliatedTo, Club_León)*, *(Salvador_Luis_Reyes, isAffiliatedTo, Club_León)*, and *(Salvador_Luis_Reyes, isAffiliatedTo, Club_Celaya)*. The first rule is also applicable as the training data contains the fact *(Pablo_Bonells, playsFor, Club_Celaya)*. We have similarly verified that the second rule for *playsFor* creates nontrivial relational paths, where the nodes are not repeated.

| Relation | Weight | Rule | MRR |
|---|---|---|---|
| *isAffiliatedTo(x,y)* | 1 | *playsFor(x,y)* | 0.463 |
| | 1 | *isAffiliatedTo(x,a) ∧ R_isAffiliatedTo(a,b) ∧ isAffiliatedTo(b,y)* | 0.582 |
| | 1 | *graduatedFrom(x,a) ∧ R_graduatedFrom(a,b) ∧ isAffiliatedTo(b,y)* | |
| | 1 | *isPoliticianOf(x,a) ∧ R_isPoliticianOf(a,b) ∧ isAffiliatedTo(b,y)* | |
| | 0.5 | *livesIn(x,a) ∧ R_livesIn(a,b) ∧ isAffiliatedTo(b,y)* | 0.585 |
| *playsFor(x,y)* | 1 | *isAffiliatedTo(x,y)* | 0.504 |
| | 1 | *playsFor(x,a) ∧ R_isAffiliatedTo(a,b) ∧ playsFor(b,y)* | 0.561 |

Table 10: Rules generated by LPRules for two relations in YAGO3-10. The MRR values for a particular rule were calculated using only the rules in the same line or above.

## A.5 ADDITIONAL EXPERIMENTAL DETAILS

Table 11 contains the list of values of the parameter $\tau$ given as input for each dataset. For larger datasets, this hyperparameter search is time-consuming, which is why we use fewer candidate $\tau$ values.

| Datasets | Values of $\tau$ |
|---|---|
| Kinship | 0.02, 0.025, 0.03, 0.035, 0.04, 0.045, 0.05, 0.055, 0.06 |
| UMLS | 0.02, 0.03, 0.04, 0.05, 0.0055, 0.06, 0.07, 0.08, 0.09, 0.1 |
| WN18RR | 0.0025, 0.003, 0.0035, 0.004, 0.0045 |
| FB15k-237 | 0.005, 0.01, 0.025, 0.05, 0.1, 0.25 |
| YAGO3-10 | 0.005, 0.01, 0.03, 0.05, 0.07 |
| DB111K-174 | 0.005, 0.01, 0.03, 0.05, 0.07 |

Table 11: Values of the parameter $\tau$ for each dataset.

Given the lists of values in Table 11, removing values from this list reduces the MRR, but not too much. But if we use entirely different values, the MRR can drop significantly. Indeed, the experiments in Table 12 show that the amount of weight given to negative sampling is very important.

Furthermore, one cannot choose the same value for different datasets, and it is best to search through a list, as we do.

| $\tau$ | Kinship | UMLS | WN18RR |
|---|---|---|---|
| 0.0001 | 0.640 | 0.560 | 0.453 |
| 0.001 | 0.686 | 0.758 | 0.461 |
| 0.01 | 0.728 | 0.799 | 0.444 |
| 0.1 | 0.667 | 0.830 | 0.385 |

Table 12: MRR obtained with fixed values of $\tau$.

We use coarse-grained parallelism in our code. For example, FB15K-237 has 237 relations, and we run the rule learning problem for each relation on a different thread. As we only have 60 cores, multiple threads are assigned to the same core by the operating system.

The run times increase significantly with increasing number of facts, and with increasing edge density in the knowledge graph. Recall that the rule learning linear program (LPR) that we solve for each relation $r$ has a number of constraints equal to $|E_r| + 1$, the number of edges labeled by relation $r$ in the knowledge graph. Given candidate rule $k$ in LPR, we need to compute $a_{ik} = C_k(X_i, Y_i)$ for each edge $i$ in $E_r$, where the $i$th edge in $E_r$ is $X_i \xrightarrow{r} Y_i$. Computing $a_{ik}$ increases superlinearly with increasing average node degree and increasing path length, and so does $\text{neg}_k$ (both calculations involve an operation similar to a BFS or DFS). See Table 13 for increasing run times on WN18RR with increasing path length. The cost of solving a linear program (LP) grows superlinearly with the number of constraints. The larger datasets (WN18RR, FB15K-237, and YAGO3-10) all have some relation which has many more associated facts/edges than the average number per relation. This leads to significant run times, both in setting up LPR for the relation and in solving LPR, in the case of FB15K-237 and YAGO3-10, and also to reduced benefits of parallelism. For example, for YAGO3-10, the relations *isAffiliatedTo* and *playsFor* have 373,783 and 321,024 associated facts, respectively, out of a total of roughly a million facts. Most relations are completed in a short amount of time, while these two relations run for a long time on a single thread each. A natural approach to reducing the run time would be to sample some of these facts while setting up LPR, but we do not do this in this work. For YAGO3-10, we do not compute $\text{neg}_k$ exactly and use sampling to obtain an approximation, as described in the main document.

One can easily parallelize the hyperparameter search process in our code. Other operations which can be parallelized are the computation of the coefficients $a_{ik}$ in LPR, but we have not done so. We thus believe there is scope for reducing our run times even further. We note that the operations we perform are not well-suited to run on GPUs.

In Table 13 we give our results for WN18RR when we use rules of length 4, 5 and 6, with the results for length 6 copied from Table 2 in the main document. The best MRR is obtained using path length 6, but takes 5 times the amount of time as path length 4. We note that the reported results for RNNLogic were obtained using a maximum rule length of 5 for WN188RR.

| Rule Length | Algorithm | MRR | H@1 | H@3 | H@10 | Time (mins) |
|---|---|---|---|---|---|---|
| 4 | LPRules | 0.449 | 0.414 | 0.465 | 0.518 | 2.3 |
| 5 | LPRules | 0.457 | 0.421 | 0.473 | 0.527 | 5.0 |
| 6 | LPRules | **0.459** | **0.422** | **0.477** | **0.532** | 11.0 |

Table 13: Results on WN18RR with maximum rule length=4,5,6 obtained using random break ranking.

## A.6 ADDITIONAL CODE DETAILS

For the experiments in this paper, in Rule Heuristic 1, we simply generate all length one and length two rules that create a relational path from the tail to head node for at least one edge $(X_i, Y_i)$

associated with relation $r$, while creating LPR for relation $r$. The number of such rules is usually small (less than 50).

In Rule Heuristic 2, for every edge $X \xrightarrow{r} Y$ in $E_r$ (associated with relation $r$), we find a shortest path, using breadth-first search, from $X_i$ to $Y_i$ in the knowledge graph $\mathcal{G}$ that does not use the edge $X \xrightarrow{r} Y$. However, when we perform column generation, we do not find a shortest path between $X_i$ and $Y_i$ for every directed edge in $E_r$. Instead, we only consider edges $i \in E_r$ that are not "implied" by the currently chosen weighted combination of rules. Such edges are indicated by large dual values, as discussed earlier. A natural improvement to this algorithm would be to find rules which create relational paths between multiple pairs of tail and head nodes $X_i$ and $Y_i$ which have large dual values.

During the search for the best $\tau$ abd $\kappa$ values, we first set up LPR or $\text{LPR}_i$ (for some $i$, when we do column generation) for a fixed value of $\tau$ and $\kappa$. Subsequently, we do not add any more rules/columns, and instead change the values of $\tau$ and $\kappa$, and evaluate the resulting LP solution on a validation set by computing the MRR, and then choosing the $\tau, \kappa$ combination which gives the best value of MRR on the validation set. During training, we obtain a weighted combination of rules for each relation separately, and then evaluate all these relations on the test set after training is complete.

All relational paths that we create (either in shortest-path calculations or during evaluation on the test set) are *simple*, i.e., they do not repeat nodes.

All our code is written in C++. The LP Solver we use is IBM CPLEX (IBM, 2019), which is a commercial MIP solver (though available for free for academic use). Any high-quality LP solver can be used instead of CPLEX, though the interface functions in our code which have CPLEX-specific functions calls would need to be changed.

### A.7 SOLUTION SPARSITY

One important feature of rule based methods is the interpretability aspect of rules. It is clear that link predictors with a very large number of rules will be harder to understand than those with few rules. In the main document, we gave the average number of rules per relation in our solutions as compared to the Neural LP solution. In our code, we vary the upper bound on complexity of chosen rules in a relation up to a certain number and use the validation set to choose the best complexity bound within the range of allowed bounds. However, we do not control the final complexity of the solution beyond the upper bound. Varying the upper bound allows us to generate solutions with lower number of rules. RNNLogic has a parameter which allows one to select the number of rules per relation used in testing; see also Figure 2a in Qu et al. (2021) where the MRR is plotted against number of selected rules. In Table 14 we give the average number of rules selected for Kinship by NeuralLP, RNNLogic and LPRules and corresponding values of MRR. These results show that LPRules is able to provide higher values of MRR for Kinship for the same number of rules selected compared to the other two codes.

| | Kinship | |
|---|---|---|
| Algorithm | MRR | S |
| NeuralLP | 0.652 | 10.2 |
| RNNLogic | 0.611 | 20.0 |
| | 0.624 | 100.0 |
| | 0.687 | 200.0 |
| LPRules | 0.739 | 11.6 |
| | 0.742 | 17.4 |
| | 0.746 | 21.0 |

Table 14: MRR and average number of rules selected (S) for Kinship.

