# OpenReview forum: "LPRules: Rule Induction in Knowledge Graphs Using Linear Programming"
_ICLR.cc/2022/Conference — ICLR 2022 Submitted_

### Official Review · Reviewer_N4fv · 2021-10-31

**Correctness:** 3
**Technical Novelty And Significance:** 3
**Empirical Novelty And Significance:** 3
**Recommendation:** 6
**Confidence:** 5

**Main Review:**

Strengths:
This manuscript exploits the idea of linear programming into knowledge graph completion tasks and then presents a model to prove the sufficiency of their idea. Some interesting experimental results are well demonstrated.

Weaknesses:
There is no in-depth comparison of the algorithm with previously known work.  Section 2 simply listed a bundle of ‘literature review’.  However, it is still hard to judge whether the algorithm is an improvement on previous work.

**Summary Of The Paper:**

To solve the problem that rule-based methods in knowledge graph completion tasks are lack of scalability to large datasets, this manuscript presents a simple linear programming model to choose rules from a list of candidate rules and assign weights accordingly. Some experiments are therefore designed and conducted to demonstrate some interesting results.

**Summary Of The Review:**

This is a well-written paper having some interesting results. A few minor revision are list below.
* The idea of using column generation techniques to add new rules is interesting. However, the methods and theoretical innovations are still not sufficient. The author(s) must dig deeper into the innovation points before the article can be accepted.
* The literature review is weak. Related work has not been investigated sufficiently before just presenting the author’s model.

---

> ### Author Response · Authors · 2021-11-19
> **Response to Reviewer N4fv**
>
> Thank you for your comments.
>
> > Weaknesses: There is no in-depth comparison of the algorithm with previously known work. Section 2 simply listed a bundle of ‘literature review’. However, it is still hard to judge whether the algorithm is an improvement on previous work.
>
> We do agree that we did not clearly explain the benefits of the new method and its position w.r.t. the literature. We have completely rewritten the related literature work to clarify the similarities and differences to other work. All the differences are in blue.

---

### Official Review · Reviewer_wtBS · 2021-11-01

**Correctness:** 3
**Technical Novelty And Significance:** 2
**Empirical Novelty And Significance:** 2
**Recommendation:** 3
**Confidence:** 5

**Main Review:**

First, the paper does have it's strong points:

S1. Simple, easy to understand approach.

S2. Only requires shortest-path computations, rule evaluation, and an LP solver.

S3. Decent results.

However, in the light of the current state-of-the-art, the paper falls short severely (W1). It's also overly limited in the type of rules that can be generated and how they can be combined (W2).

W1. Does not include SOTA approaches and falls behind in all areas where benefits are claimed: [A] (not cited) proposes a rule learner which (1) has a larger language bias, (2) is orders of magnitude faster than the proposed approach on a less powerful machine, and (3) produces higher-quality predictions, and (4) is simple. E.g., on the largest YAGO3 dataset: paper 21h, .45 MRR, 60 cores; [A] 16min, >=.49 MRR, 4 cores. It's not clear what the benefits of the proposed approach really are, if any.

[A] Christian Meilicke, Melisachew Wudage Chekol, Daniel Ruffinelli and Heiner Stuckenschmidt: Anytime Bottom-Up Rule Learning for Knowledge Graph Completion. IJCAI 2019.

W2. The rule generation heuristics are well-known; there is no novelty here (it's also not claimed). It appears that only chain rules that correspond to shortest path can be learned. The language bias is thus quite limited and does not include other types of rules known to be usefule, most notably, rules involving constants ("X lives in New York" => "X speaks English"). There has been prior research in how rules are combined (e.g., [A] uses the most confident rule); it's not clear whether a linear combination of rules is a good idea in the first place. Here an ablation and comparison to prior approaches is missing. As for pruning negative-weight rules: There isn't much pruning at all: only 50 candidate rules are generated per relation in the first place in the experimental study. These candidates are useful in that they are generated sequentially based on the candidates so far, but this idea is also present in prior work (e.g, RNNLogic, [A]).


**Summary Of The Paper:**

The paper proposes a simple method to generate logic rules, where rules are generated by a shortest-path heuristic and rule weights by solving a linear program.


**Summary Of The Review:**

In the light of the current state-of-the-art, the paper falls short severely (W1). It's also overly limited in the type of rules that can be generated and how they can be combined (W2).

---

> ### Author Response · Authors · 2021-11-19
> **Response to Reviewer wtBS**
>
> We thank the referee for reading our paper and pointing out the reference [A] and code AnyBURL. We discuss these in the revision. All changes are in blue.
>
> > It's also overly limited in the type of rules that can be generated and how they can be combined (W2).
>
> We address both parts of this comment below.
>
> > Limited in the type of rules
>
> We have focused on entity-independent chain-like first-order rules that correspond to relational paths as they are still highly desirable.
> 1. They are entity independent and thus well-suited to inductive settings when one deals with new entities not seen before.
> 2. They are very compact representations of knowledge, especially when few rules are chosen. They are thus easy to interpret and understand.
> 3. It is partly because of this that many papers/codes follow this paradigm: NTP, NeuralLP, DRUM, MINERVA, RNNLogic, PathRank, M-Walk etc.
> Another referee points out NLIL (which generates more general rules than ours) and GRAIL (which does subgraph reasoning), and we cite them now. But we stand by the value of studying the problem of entity-independent first-order logic rules (we will discuss this more later).
> 4. We do now extensively cite AnyBURL and compare with it. We have accordingly rewritten the "Related Work"
>
> >  how they can be combined (W2).
>
> This is a valid point that there are different ways. We now give some explanation in the related work section and when we introduce the model.
>
> > However, in the light of the current state-of-the-art, the paper falls short severely (W1).
> > [A] (not cited) proposes a rule learner which (1) has a larger language bias, (2) is orders of magnitude faster than the proposed approach on a less powerful machine, and (3) produces higher-quality predictions, and (4) is simple. E.g., on the largest YAGO3 dataset: paper 21h, .45 MRR, 60 cores; [A] 16min, >=.49 MRR, 4 cores. It's not clear what the benefits of the proposed approach really are, if any.
>
> We strongly disagree with this assessment. We agree with points (1)-(4), and we have extensively discussed AnyBURL in the revised paper. But our disagreement stems from the fact that AnyBURL, though a rule-learner, is closer to embedding-based methods in that it produces:
> 1.  entity-dependent rules ("the language bias" the referee mentions) which cannot be used in an inductive setting.
> 2. vast numbers of rules rendering the rule set entirely noninterpretable.
> 3. over 12,000 rules per relation in YAGO3-10 versus 7.8 for LPrules, when we run both codes on our machine (see Table 5). This number (12,000) is consistent with Figure 2 in the AnyBURL paper.
> 4. Many rules with grounded values instead of few rules with purely existential variables. For example, for YAGO3-10, while we learn the rule is_affiliated_to(x, y) <- plays_for(x,y) for players x and clubs y (x and y are variables here), AnyBURL learns thousands of rules of the form  is_affiliated_to(a, y) <- plays_for(a,y) where 'a' is a specific player, and y is a variable. It is clear that the single rule we give has way more interpretability and human readability. In theory, the rule we learn lies in the space searched by AnyBURL (the space of "cyclic" rules), but the specific implementation used does not return our rule, likely because of the greedy approach used.
> 5. In our view, the fact that AnyBURL does not prune rules is a serious weakness.
>
> > it's not clear whether a linear combination of rules is a good idea in the first place.
>
> We agree that there is no guarantee that this is the best option. But it is important to note that there is a lot of literature on using linear combinations of rules, and we try to improve on this literature. Finally, we also note that the maximum confidence idea in AnyBURL leads to a vector-valued score (with scores sorted in the vector) for each candidate answer entity for a query (t, r, ?), and one needs to compare two vectors lexicographically to determine the better entity, making human use and analysis of AnyBURL even more challenging.
>
> > There isn't much pruning at all: only 50 candidate rules are generated per relation in the first place in the experimental study.
>
> The 50 candidates is only for YAGO3-10. It is in the low hundreds for all the other problems. We clarify this latter fact in the text in the paper just before the conclusion.
>
> > These candidates are useful in that they are generated sequentially based on the candidates so far, but this idea is also present in prior work (e.g, RNNLogic, [A]).
>
> Indeed this idea is present, but our model is different and it is important to show that we can repeat this idea in our model.
>
> ** Summary
>
> Our code is broadly comparable to codes such as RNNLogic, NeuralLP, DRUM etc. and its contribution should be assessed in relation to these papers. Our metrics are scalability, compactness of rule sets, prediction performance.

---

> > ### Comment · Reviewer_wtBS · 2021-11-25
> > **Comments on author feedback**
> >
> > The author feedback does not change my assessment. It makes me question, in fact, whether the authors value the publication of their paper higher than a careful and balanced consideration of prior work.
> >
> > My first major concern was that the types of produced rules are limited (only paths, no constants). The authors argue:
> >
> > (1) Using constants in rules is a bad idea because rules with constants ("entity-dependent" in paper) are not useful in inductive settings. I disagree. For example, a rule such as "speaks(x,English)<-bornIn(x,England)" is very useful in inductive settings. Avoiding such rules comes at a significant loss in modeling power and may explain the lower overall prediction performance of the paper's approach compared to prior work.
> >
> > (2) Only few rules make the rule set easy to interpret and understand. I'd argue that having many rules, even millions of rules, is not problem as long as one can pinpoint a few rules that led to each prediction (i.e., a local explanation). A max aggregation of scores such as the one performed by AnyBURL allows to output the precise single rule that produced each prediction, so results are interpretable.
> >
> > (3) Other papers did it. I am not sure what the argument here is.
> >
> > My second major concern was that the proposed method falls behind AnyBURL, which appears to be more powerful, faster, higher quality, and simpler. The authors agree with this statement, but disagree with me that this has implications to their work. Arguments 1+2 are as above (which see). Argument 3-5 are that AnyBURL produces many more rules due it's use of constants and that some of those rules are useless. I do agree that some rules are useless (like the example given by the authors) and that this is a weakness (but, as argued above, other such rules are also useful). Be that as it may, an argument for small rule sets can be made: it provides a global explanation (since it's feasible to look at all rules) and a small model size.
> >
> > This does not immediately imply that the paper's approach is the right way to go though. I've contacted the authors of AnyBURL asking whether it's possible to disable the generation of certain types of rules (most notably, those involving constants), also pointing them to this OpenReview page. It is indeed possible (via the configuration file) and, after my inquiry, now additionally documented on the AnyBURL homepage. (They also said that AnyBURL reaches a small ruleset and comparable performance in 10 seconds runtime. This is not a peer-reviewed result, though, so let's ignore it.) My point is that the authors did not consider such baselines at all, and they also did not consider other baselines such as post-processing the results of a super-fast miner such as AnyBURL.
> >
> > Finally, the authors state that their work should be assessed in relation to prior work that "falls behind" (my words), but not against prior work that leaps ahead (also my words). Why, exactly? They then state that their metrics are scalability (AnyBURL clearly ahead), compactness of rule sets (not explored), prediction performance (AnyBURL clearly ahead): even using the author's suggested metrics, the position of this work is unclear at best.
> >
> > Overall, the paper fails to make convincing arguments to be further considered at this ICLR conference.
> >
> > Other points:
> >
> > The authors argue that max aggregation leads to a vector-valued score. It does not: The score is the confidence of the highest-confidence rule producing the result. One score, one rule.
> >
> > The revised paper states "Henceforth, rule-based codes does not refer to AnyBURL." The argument is that an approach that can (but also may not, see above) produce rules involving constants is not rule-based. This is not convincing.

---

> > > ### Author Response · Authors · 2021-12-01
> > > **Response to "Comments on Author Feedback"**
> > >
> > > > (1) Using constants in rules is a bad idea because rules with constants ("entity-dependent" in paper) are not useful in inductive settings. I disagree. For example, a rule such as "speaks(x,English)<-bornIn(x,England)" is very useful in inductive settings. Avoiding such rules comes at a significant loss in modeling power and may explain the lower overall prediction performance of the paper's approach compared to prior work.
> > >
> > > We will concede that entity-dependent rules can be useful and should be considered in a practical setting. A perfect setting would be say medical diagnosis, where ignoring the individual patients identity would likely give worse quality results. Furthermore, we agree that not all entities in a knowledge graph are of the same type. In YAGO3-10, there are entities for players, and for national teams. So a rule involving an unknown player x and say the English team, would be useful, and could be used in an inductive setting where new unknown players are considered.
> > >
> > > However, there are many settings where one wants rules which do not involve the identity of an individual. To go back to the medical setting, a doctor may care for a patients identity, but a public health professional will definitely not and would like generally applicable rules, e.g., high_cholesterol(x) and child_of(x,y) and heart_disease(y) => heart_disease(x). There are numerous other situations where entity-independent rules are desirable. This is why we cannot agree with the referee dismissing this body of work (on entity-independent rules) as out of date. If modeling power were the only criteria, then embedding based methods would suffice for KG link completion.
> > >
> > > > (2) I'd argue that having many rules, even millions of rules, is not a problem as long as one can pinpoint a few rules that led to each prediction (i.e., a local explanation). A max aggregation of scores such as the one performed by AnyBURL allows to output the precise single rule that produced each prediction, so results are interpretable.
> > >
> > > - Our first response is that the referee chooses one model of interpretability (local explanations). But there are numerous papers arguing for the value of the other model (global explanations). Our point is that it is not simply our work, but a significant body of literature that associates interpretability with "global" interpretability. We can definitely add pointers to this literature, but we cannot agree that it can be trivially dismissed. See below.
> > >
> > > 1. Himabindu Lakkaraju, Stephen H. Bach, and Jure Leskovec. Interpretable decision sets: A joint framework for description and prediction. In Proc. ACM SIGKDD Int. Conf. Knowl. Disc. Data Mining (KDD), pages 1675–1684, 2016.
> > > 2. Sanjeeb Dash, Oktay Günlük, and Dennis Wei. Boolean decision rules via column generation. In Advances in Neural Information Processing Systems, pp. 4655–4665, 2018.
> > > 3. Tong Wang, Cynthia Rudin, Finale Doshi-Velez, Yimin Liu, Erica Klampfl, and Perry MacNeille. A Bayesian framework for learning rule sets for interpretable classification. Journal of Machine Learning Research, 18(70):1–37, 2017.
> > >
> > > In fact, the second paper won an interpretable machine learning challenge from FICO (see https://community.fico.com/s/explainable-machine-learning-challenge) partly because of the compact rules generated in the paper on FICO's dataset.
> > >
> > > - The only purpose of rules is not prediction; at times rules are converted into actual actions. Again going back to the public health example, one might convert rules of the type we gave into monitoring actions.
> > >
> > > - Rules also lead to insight and abstraction; at times abstraction is the explicit goal even at the cost of predictive performance. In this case and in the previous one, the actual size of the rule set matters.
> > >
> > > - We note that the total number of rules produced by AnyBURL is close to half a million, and close to half the number of facts in YAGO3-10. For Kinship and UMLS, the number of rules even exceeds the number of facts. The point here is not to criticize AnyBURL (and we do not do so in our modified version of the paper), but to point out that there are many situations where this is simply not desirable.
> > >
> > > > (3) Other papers did it. I am not sure what the argument here is.
> > >
> > > The argument is simple. There is a huge body of literature on entity-independent rules (though we only gave 10-15 papers in our work).

---

> > > > ### Author Response · Authors · 2021-12-01
> > > > **Response continued**
> > > >
> > > > >  (They also said that AnyBURL reaches a small ruleset and comparable performance in 10 seconds runtime. This is not a peer-reviewed result, though, so let's ignore it.) My point is that the authors did not consider such baselines at all
> > > >
> > > > We do not deny this could be possible, but if the original paper does not talk about this as a goal, it cannot be expected that readers should check if getting small rule sets is possible out of AnyBURL. Just as one cannot expect readers of our paper to expect our code to generate entity-dependent rules.
> > > >
> > > >  And even if AnyBURL can do it, the observation that compact rule sets can give good performance is an original claim in our paper for the datasets mentioned. We are not aware of many other papers that attempted to achieve this as an explicit goal.
> > > >
> > > > > they also did not consider other baselines such as post-processing the results of a super-fast miner such as AnyBURL.
> > > >
> > > > - We have done some limited experiments with post-processing, and can verify that the MRR for FB15K-237 drops to 0.210 from 0.278 after removing entity-dependent rules from the 100 second run of AnyBURL, though the drop is less for WN18RR (from 0.479 to 0.453)
> > > >
> > > > > The revised paper states "Henceforth, rule-based codes does not refer to AnyBURL." The argument is that an approach that can (but also may not, see above) produce rules involving constants is not rule-based. This is not convincing.
> > > >
> > > > Fair point. It would simply have been better to refer to codes other than AnyBURL as "entity-independent rule-based codes".
> > > >
> > > > > The authors argue that max aggregation leads to a vector-valued score. It does not: The score is the confidence of the highest-confidence rule producing the result. One score, one rule.
> > > >
> > > > This is only partially correct. The authors of AnyBURL say: "We apply a rather simple but efficient approach. We order the candidates via the maximum of the confidences of all rules that have generated the candidates. If the maximum score of several candidates is the same, we order these candidates via the second best rule that generates them, and so on, until we find a rule that makes a difference."
> > > >
> > > > Here the candidates refer to possible solutions to the query (a, r, ?). The algorithm defined above (looking at the second best rule and the third best rule etc. to distinguish between candidates is exactly the same as assigning a vector-valued score, where the vectors have different lengths, and values in the vector decrease with increasing index, and candidates are compared by comparing vector-scores lexicographically.
> > > >
> > > > > but not against prior work that leaps ahead (also my words). Why, exactly?
> > > >
> > > > We do not say this. Which is why we included AnyBURL in our results. But we do not think that the published results are comparable.
> > > >
> > > >  > They then state that their metrics are scalability (AnyBURL clearly ahead), compactness of rule sets (not explored), prediction performance (AnyBURL clearly ahead): even using the author's suggested metrics, the position of this work is unclear at best.
> > > >
> > > > We just noticed that AnyBURL separates out the process of generating rules from evaluating them. So the 100 seconds (which we report for AnyBURL) is not comparable to our time. We report in our running times all the times including evaluation time (other than for AnyBURL). Once that is done, AnyBURL goes up to 15 minutes or more on our multi-core machine. We have explained in our paper (before the revision) that for problems such as YAGO3-10, our code essentially becomes sequential as we run one relation on one core, and YAGO3-10 has one dominant relation. The point is that with a more fine grained parallelization, we could easily speed up our code significantly.
> > > >
> > > > > Overall, the paper fails to make convincing arguments to be further considered at this ICLR conference.
> > > >
> > > > We should probably have justified better the need for compact rule sets in the paper (as we tried to do in the response above). We feel it is not fair to rule out our paper by simply saying: 1) the prior literature on entity-independent rules is not relevant any more; 2) prior literature on the value of compact rules is not relevant any more; 3) AnyBURL can do many of these things (compact rule sets + entity-independent rules only) and get good results even if the authors never talked about this in their paper.

---

> > > > > ### Comment · Reviewer_wtBS · 2021-12-02
> > > > > **Comments on 'Response to "Comments on Authors Feedback"'**
> > > > >
> > > > > Thanks for the response! Looks like we are slowly converging.
> > > > >
> > > > > Generally, I do think that this line of work can be valuable, but the present paper is not there yet. I hope that the feedback from the ICLR reviews and subsequent discussion give helpful directions.
> > > > >
> > > > > Generally, quite some responses argue against points I did not actually make (most notably, points 1+2 in the bottom line conclusion). Quick thoughts:
> > > > >
> > > > > I did not argue that entity-independent rules are out of date or uninteresting, I said that they lead to a significant loss in modeling power. And this they do, as your experiments also indicate.
> > > > >
> > > > > The stated example of "heart_disease(y) => heart_disease(x)" uses unary predicates. In knowledge graphs, such information is represented via properties of form "has_disease(x, heart_disease)", the stated rule is equivalent to "has_disease(y, heart) => has_disease(x, heart)", a rule with constants. If the argument of the authors is that certain constants (e.g., certain named entities) should not appear in inductive rules, whereas others are fine (e.g., types or constants such as heart), I'd agree. Note that the constants which are allowed to appear can be used to induce unary predicates (e.g., "heart_disease(x) <= has_disease(x, "heart")"), just as the authors did in their response (but not in the paper/experimental study).
> > > > >
> > > > > I did not argue against global explanations. My point was that local explanations are possible with many rules and max-aggregation. As I originally stated: "an argument for small rule sets can be made: it provides a global explanation (since it's feasible to look at all rules) and a small model size". We actually agree on this point. The paper doesn't spell it out though.
> > > > >
> > > > > The authors acknowledge that it's indeed possible to get compact rule sets with existing methods (in the case of AnyBURL, by setting a configuration flag). The benefits/drawbacks of the proposed method for producing compact rule sets is in question as is which language bias one should have; it's not in question whether it's worth having compact rule sets.
> > > > >
> > > > > Finally, it can be expected that the authors due their due diligence and consider related work as well as simple baselines appropriately.

---

### Official Review · Reviewer_3BGR · 2021-11-03

**Correctness:** 3
**Technical Novelty And Significance:** 3
**Empirical Novelty And Significance:** 3
**Recommendation:** 6
**Confidence:** 3

**Main Review:**

Generally, the paper is well written, proposes a novel approach to obtain a scoring function given a set of rules and a KG, and has convincing experimental results.

The only criticism I’d have is some missing related work. The authors cite exhaustively from the literature on knowledge graph embedding methods and rule-based approaches. However, there is a large body of work in the statistical relational learning community, focusing precisely on computing a scoring function (here: marginal probability or MAP state; and/or computing rule weights) in large probabilistic models defined through weighted rules. Below are two representative examples who proposed (among other things) column generation techniques and formulating the problem as an ILP. It’s not an LP but one could relax the ILP. I believe this is work that should be cited as related. In fact, I believe a discussion of these related approaches in the context of SRL might be illuminating and bring SRL researcher to pay attention to the idea proposed here in the context of knowledge graph completion.

https://arxiv.org/abs/1206.3282

https://arxiv.org/abs/1304.4379

**Summary Of The Paper:**

The paper presents a method to obtain weights for a knowledge graph scoring model for link prediction. There are numerous good algorithms for mining rules from knowledge graphs (e.g. AnyBURL and AMIE). Less research has focused on the problem of creating scoring functions based on an (implicit) list of rules. This is what the present paper proposes.

The core idea is to formulate a linear program whose solution corresponds to a scoring method. Instead of incorporating all rules (there are typically many!) into the LP formulation, the authors propose a column generation approach.



**Summary Of The Review:**

Good paper but some discussion of highly related work is missing

---

> ### Author Response · Authors · 2021-11-22
> **Response to Reviewer 3BGR**
>
> Thank you for the careful review, and suggested citations. We will add these citations by tomorrow's deadline.
>
> >  large body of work in the statistical relational learning community, focusing precisely on computing a scoring function
>
> We agree and we have tried to rewrite the introduction to mention this more explicitly in the limited space we have.
>
> >  Below are two representative examples who proposed (among other things) column generation techniques and formulating the problem as an ILP
>
> We have looked at these papers carefully, and we have cited them (see the new related work section). However, the specific problem they solve seems related but not identical. If one takes the dual of the LP problem we solve, then one has exponentially many rows, and a cutting-plane/constraint generation approach would be necessary, and one would generate violated constraints on the fly and them to current set of constraints. In the two papers you suggest, the authors work with exponentially many constraints. We now mention this similarity. However, it seems to us that beyond this the problems are different unlike the papers by Kok and Domingos and by Lao and Cohen that we cite. The latter two explicitly find parameters of a scoring function (i.e. rule weights). The ones you suggest, find settings of predicate values that would lead to observed evidence, and this setting leads to intrinsically binary problems.

---

> > ### Comment · Reviewer_3BGR · 2021-12-02
> > **Post rebuttal statement**
> >
> > I've read the other reviews, the rebuttals, and the discussion between the authors and reviewer wtBS.
> >
> > A couple of comments: I find the criticism of reviewer wtBS and especially the score of 1 too harsh. The focus of said reviewer seems to be on the ability of new methods to "beat" existing methods and if that's not the case, new ideas are automatically disqualified. As mentioned before, I like the ideas presented in the submitted work and even if some might agree that AnyBURL's way to select top rules is a better method to explain the predictions, there is room for various competing/alternative methods. If I've learned one thing from the existing XAI literature, it's that there is no one-fits-all solution. The score of 1 is also harsh given that the method is technically "correct".
> >
> > Now, I would argue that a lower score than a 8 is justified due to the missing related work and the rosy positioning of the new method in comparison to existing methods such as AnyBURL. Hence, I'll lower my score to a 6 (marginally above acceptance). I would suggest the authors to continue their line of work and to take the input of the reviewers into account while not being discouraged.

---

### Official Review · Reviewer_PFXs · 2021-11-03

**Correctness:** 3
**Technical Novelty And Significance:** 2
**Empirical Novelty And Significance:** 2
**Recommendation:** 3
**Confidence:** 3

**Main Review:**

Advantages:
* The problem of learning logical rules is very crucial.
* The proposed model is very simple and seems to be effective.
* The efficiency is better than some previous works.

Disadvantages:
* The paper is hard to follow, which requires improvements.
* The motivation of the model is confusing, which seems to have no significant improvement compared to existing works such as Neurallp and DRUM. They reduce the search space by changing the order of summation and multiplication when calculating rules, which can also be regarded as low-rank decomposition of rule-score tensors.
* In my opinion, the most time-consuming module of Neurallp and DRUM is LSTM. If they directly use the coefficients of relations in the rule as the learnable parameters, it would be much faster and may achieve comparable performance. I am not sure the proposed method is also faster than the simplified version of Neurallp and DRUM.
* There are several works that the author didn’t compare with, such as [1,2], which can learn more complex rules.

Minor issues:
I -> i in equation 2.

[1] Yang, Yuan, and Le Song. "Learn to Explain Efficiently via Neural Logic Inductive Learning." International Conference on Learning Representations. 2019.

[2] Teru, Komal, Etienne Denis, and Will Hamilton. "Inductive relation prediction by subgraph reasoning." International Conference on Machine Learning. PMLR, 2020.



**Summary Of The Paper:**

This paper proposed a simple linear programming model for learning logical rules for KG completion. The model selects candidate rules from KG with explicit constraints and then solves a linear programming problem. The authors conduct experiments on several public datasets and the model has better efficiency than baseline models.

**Summary Of The Review:**

In summary, the paper presentation requires improvements and the motivation for designing such a model is not clear. There are several methods that the authors did not use for comparison.

---

### Decision · Program_Chairs · 2022-01-20

**Decision:**

Reject

**Comment:**

The paper shows how to make use of a linear program for extracting logical rules for knowledge graph completion. Overall, the reviewers and I agree that this is an interesting and important direction for research. Moreover, the presented approach shows good performance with rather small sets of rules extracted. However, all reviewers point out that the related work is not well discussed. While the authors have improved the related work sections during the rolling discussion, overall the positioning of the new method has still to be improved, including a better empirical comparison across different datasets. Overall, we would like to encourage the authors to polish their line of research based on the feedback from the reviews.